# Investigating Eating Habits of Children Aged between 6 Months and 3 Years in the Provinces of Modena and Reggio Emilia: Is Our Kids’ Diet Sustainable for Their and the Planet’s Health?

**DOI:** 10.3390/healthcare12040453

**Published:** 2024-02-10

**Authors:** Lucia Palandri, Laura Rocca, Maria Rosaria Scasserra, Giacomo Pietro Vigezzi, Anna Odone, Lorenzo Iughetti, Laura Lucaccioni, Elena Righi

**Affiliations:** 1Section of Public Health, Department of Biomedical, Metabolic and Neural Sciences, University of Modena and Reggio Emilia, 41125 Modena, Italy; marierose@inwind.it (M.R.S.); elena.righi@unimore.it (E.R.); 2Clinical and Experimental Medicine PhD Program, University of Modena and Reggio Emilia, 41125 Modena, Italy; 3Pediatric Unit, Department of Medical and Surgical Sciences of the Mother, Children and Adults, University of Modena and Reggio Emilia, 41125 Modena, Italy; laurarocca28@gmail.com (L.R.); lorenzo.iughetti@unimore.it (L.I.); llucaccioni@unimore.it (L.L.); 4Department of Public Health, Experimental and Forensic Medicine, University of Pavia, 27100 Pavia, Italy; gvigezzi@hotmail.com (G.P.V.); anna.odone@unipv.it (A.O.); 5Collegio Ca’ della Paglia, Fondazione Ghislieri, 27100 Pavia, Italy; 6Neonatology Unit, Department of Medical and Surgical Sciences of the Mother, Children and Adults, University of Modena and Reggio Emilia, 41125 Modena, Italy

**Keywords:** infant nutrition, dietary habits, sustainability, health promotion, greenhouse gas emissions, obesity prevention, public health, school meals

## Abstract

A healthy and balanced diet is crucial for children’s well-being and aids in preventing diet-related illnesses. Furthermore, unhealthy dietary habits indirectly impact children’s health, as the food industry stands as one of the primary drivers of climate change. Evidence shows the Mediterranean diet is sustainable for both children’s and the planet’s health. The aim of this cross-sectional study was to evaluate the eating habits of children aged between 6 months and 3 years, in the province of Modena and Reggio Emilia, in Italy, along with their adherence to the guidelines for a healthy diet, and examine the role of pediatricians in promoting knowledge about nutrition and sustainability. In our sample (218 children), most children exceeded the recommended meat and cheese intake, while consuming insufficient amounts of vegetables, fruit, and legumes. Vegetable and fruit consumption declined with the increase in age category while eating sweets, soft drinks, and processed food increased. Incorporating school meals’ data into this analysis, we observed a modification in dietary compliance, characterized by an increase in meat and cheese consumption, alongside improvements in the intake of vegetables, fruits, fish, eggs, and legumes. This study suggests that supporting an integrated approach that combines social and educational initiatives is crucial. Future research should prioritize fostering sustainable eating habits within communities to facilitate dietary habits’ transformation and encourage healthier lifestyles.

## 1. Introduction

The first three years of life are crucial for establishing healthy and balanced dietary habits to ensure that infants receive all the necessary nutrients for proper physical and cognitive development. This is vital in preventing nutrition-related problems, particularly obesity [1].

Early in life, it is recommended to adopt a balanced diet with appropriate fat and protein intake and ensuring proper consumption of fruit and vegetables. Inadequate food items consumption, imbalances in macronutrients, as well as genetic and socio-economic factors, can contribute to an obesogenic environment. Pediatric obesity has seen a significant increase in prevalence over the last few decades, especially in high-income nations [2].

Indeed, the period of life between 6 and 36 months is characterized by a rapid transition from a milk-based diet to the introduction of various food categories daily. From the outset, parents should be able to choose healthy foods for the complementary feeding period. It is important to highlight that unbalanced pediatric nutrition and unhealthy dietary habits present two significant criticalities.

On one hand, pediatric nutrition stands as a pivotal life-course health determinant, notably influencing diseases like obesity, diabetes, and cardiovascular ailments [3]. These conditions incur substantial healthcare costs and societal burdens while impacting individual life expectancy and quality [4,5]. Already in 2015, Gorski and Roberto suggested how unhealthy dietary patterns, such as low fruit consumption and high levels of sodium intake, were responsible for up to 10% of global disability-adjusted life years [6]. Overweight and obesity, which represent important risk factors for diseases such as type 2 diabetes, heart disease, and high blood pressure, contribute significantly to mortality and disabilities in Europe, accounting for over 1.2 million deaths annually, approximately 13% of the region’s total mortality. Childhood obesity not only affects immediate health and education but also predicts adult obesity and elevates the risk of non-communicable diseases. Italy’s ‘Okkio alla Salute’ national survey (2019) [7] revealed concerning rates: 20.4% overweight and 9.4% obese children aged 8–9, with slightly higher obesity rates in males. Geographically, southern regions show a higher prevalence. Socioeconomic disparities and shorter breastfeeding durations correlate with increased obesity rates [8]. Numerous studies observe a global obesity trend linked to an increasing preference for Western diets, abandoning traditional ones. The initial decline in Mediterranean diet adherence, circa the 1960s, coincided with economic growth and food market industrialization, promoting processed foods and the greater availability of red meats and sugars, reducing vegetables and cereals consumption. Presently, Westernized diets, especially among young generations, contribute to lifestyle imbalances impacting energy intake and expenditure [3,9,10].

On the other hand, unhealthy dietary habits also indirectly impact children’s health. The food industry stands as one of the primary drivers of climate change. With current diets and production practices, feeding 8 billion people is degrading terrestrial and aquatic ecosystems, depleting water resources, and contributing to greenhouse gas emissions [11]. The food system impacts the environment in three primary ways: CO2 emissions, water consumption, and land use, further affecting biodiversity changes and water eutrophication [12].

Nutrition significantly influences disease prevention, impacting human health in morbidity and mortality. Additionally, as observed earlier, the food system notably impacts climate change, affecting agricultural productivity and generating adverse health outcomes. The Lancet Commission in 2019 categorized global epidemics of malnutrition, obesity, and climate change as components of a global syndemic, coexisting and interacting to produce intricate consequences [13]. Among all food categories, meat (especially red meat), dairy products, sweet beverages, and ultra-processed food represent the principal drivers of this global syndemic.

A healthy and balanced diet is crucial for the physical and mental well-being of children and aids in preventing diet-related illnesses. Evidence shows the Mediterranean diet as beneficial for both mental and physical health in adults and children [14]. This diet comprises a high consumption of vegetables, fruit, nuts, whole grains, and olive oil, moderate intake of fish and poultry, and low consumption of sweets, red meat, and dairy [15].

The aim of this study is to evaluate the eating habits of children aged 6 months to 3 years living in the province of Modena and Reggio Emilia, in Italy. The goal is to assess food items’ consumption frequency and compare intake to pediatric dietary guidelines, accounting for school meals offerings as well, as there is a lack of information about eating habits in this specific age range. The secondary aim of this study is also to assess parents’ knowledge regarding the role of family pediatricians in disseminating knowledge about a healthy diet and the environmental impact of nutrition.

## 2. Materials and Methods

### 2.1. Study Design

This cross-sectional study was conducted through the development of a questionnaire targeting the parents of children aged between 6 months and 3 years. Adherence to nutritional guidelines was assessed using age-appropriate proportions from the 2018 CREA-Healthy Eating Guidelines [16]. We investigated items, portion, and frequencies and aggregated into 14 groups according to the CREA guidelines: *Total Cereals*: pasta, bread, rice, other cereals (barley, spelt, corn, quinoa, buckwheat, oat, kamut), bread substitutes (crackers, breadsticks), breakfast cereals. *Total Meat*: red meat (beef, veal, pork, sheep, goat, game meat, red meat in the pasta sauce), white meat (chicken, turkey, rabbit, white meat in the pasta sauce). *Processed Meat*: cured meat, dried meat, cooked ham, salami, bresaola, sausage, industrial burgers. *Fish*: fresh fish, frozen fish, clams, crustaceans. *Legumes*: beans, peas, chickpeas, lentils, grass peas. *Total Cheese*: light cheese, with less than 25% fat (mozzarella, stracchino, crescenza, ricotta, provola); fatty cheese, with more than 25% fat (robiola, caciotta cheese, aged cheese). *Vegetables:* fresh vegetables, leafy salads (excluding potatoes). *Sweets*: ice cream, pudding, candies, Nutella, chocolate, bakery products (cake, tarts, croissants, biscuits). *Sweet Soft Drinks*: fruit juice, orange soda, sweet tea, coke soda. *Processed Food*: sweet snacks, savory snacks, potato chips, ready-to-eat and pre-cooked food, fish sticks, fast food, fruit in syrup. To help parents to correctly assess portions, age-specific food portion images drawn from the photographic atlas of food portions for the pediatric age—Bassani Scotti Institute [17]—were shown in the questionnaire.

The questionnaire was designed to be easily accessible and anonymously fillable. The study did not gather specific personal data or identifiable information (e.g., names). We drew insights and utilized materials from the scientific literature related to studies on adherence to the Mediterranean diet in both adults and children [18,19].

The questionnaire was designed using Google Forms to facilitate its prompt online distribution and to ensure compatibility with smartphones, allowing completion within approximately ten minutes. It was distributed from 7 February to 11 May 2023, through various channels, including social media platforms targeting parents, instant messaging, and QR codes on flyers distributed to parents in Modena Hospital’s pediatric clinics or displayed on posters in pediatricians’ waiting areas.

The questionnaire was subdivided into three sections: *General socio-economic data*, aimed at characterizing the study population by collecting personal information from parents, detailing their child’s and family’s status; *Eating Habits*, designed to investigate the dietary habits of our pediatric population, estimating the quantities of food items consumed on a daily or weekly basis: parents were asked to respond only regarding meals that the child consumed at home; *Role of the Pediatrician*, which aimed to explore how frequently pediatricians provide recommendations on proper nutrition and the environmental sustainability of food production (details of variable categories in Table 1 and Table 2).

### 2.2. Processing of the Results

Working groups were defined according to the European socioeconomic groups (ESeG) [20] classification composed of 3 classes of 9 categories: upper class (managers, professionals), middle class (technicians and associated professionals employees, small entrepreneur), working class (clerks and skilled service employees, skilled industrial employees, lower-status employees), and not working.

For dietary assessment, after collecting responses for single food items, we aggregated the categories ‘*Total cereals*’, ‘*Total meat*’, and ‘*Total cheese*’, derived from the overall consumption frequencies of their respective subclasses. Furthermore, to provide a more accurate assessment, the sample was divided into those who ate lunch at school and those who did not. In the first scenario, we supplemented parent-provided food with item frequencies from school meals. To ensure anonymity, we could not ascertain the child’s school nor their exact food consumption, we examined school menus from several towns in the province of Modena and Reggio Emilia to determine the average frequency of offerings for different food categories, with the assumption that the child consumed all provided items. In these provinces, school meals are strictly regulated by the two local health authorities and are similar across different educational facilities.

We then categorized responses about food consumption frequency, labeling them as “According to Guidelines”, “Below Guidelines”, or “Above Guidelines”. Yet, for certain foods (cereals, fresh fruit, and vegetables), excess consumption did not indicate non-compliance with guidelines. In these cases, a higher intake was marked “According to Guidelines”. To gauge overall adherence, we assigned scores: 1 for “According to Guidelines” and 0 for “Below or Above Guidelines”. This scoring system allowed us to tally scores across food categories, generating a range from 0 to 14, indicating adherence levels to Italian dietary guidelines. For instance, a higher score suggests closer alignment with recommended consumption frequencies outlined in dietary guidelines. This method facilitated an overview of our population’s adherence to dietary recommendations, emphasizing variations in adherence levels to these guidelines among the studied group. The study aimed to assess our sample’s adherence to the Italian dietary guidelines rather than the Mediterranean diet. Consequently, previously validated Mediterranean diet indices were not utilized. Additionally, the most commonly used scores in the pediatric population, such as the relative Mediterranean diet score (rMED) [21,22] or the KIDMED score [19,23,24], have been validated from 2 years of age, which does not encompass our specific population.

Finally, to assess if there were patterns in dietary adherence, we recategorized adherence responses as yes (According to Guidelines) and no (Below or Above Guidelines), and we performed an adherence clustering analysis.

### 2.3. Statistical Analysis

Categorical variables were summarized by absolute and relative frequencies. The mean and standard deviation (SD) or median and interquartile range (IQR) were used to summarize continuous variables according to their distribution. Pearson’s chi-square test or Fisher’s exact test was used to compare categorical variables.

For the analysis of adherence clustering, we used the Jaccard similarity coefficient (values between 0 and 1, where the higher the number, the higher the co-occurrence of adherence to a specific food category) and presented as a hierarchically ordered heatmap. Clustering was obtained using the complete linkage agglomerative hierarchical clustering method. Estimation of the optimal number of clusters was performed using the silhouette method (Appendix A) [25]. Missing data were dealt with in complete-case data analysis. We used Excel (Microsoft, Redmond, WA, USA) and R (R Core Team version 4.1.1 (2021), Vienna, Austria [26]) for descriptive statistics, plotting and advanced statistical analysis, with packages including tidyverse, ggplot2, ggrepel, dendextend, gplots, and UpSetR.

### 2.4. Ethical Considerations, Patient Information, and Written Informed Consent

This study did not require the approval of an ethics committee because the questionnaire data were anonymous, making it impossible to identify and harm any respondent. Moreover, neither drugs nor medical devices were prescribed or administered. As a result, the responses were collectively examined while taking into account Italian and European regulations governing the management of personal data [27,28,29].

In more detail, anonymity was granted by not requesting the name, surname, and date of birth. In order to retrieve information about the geographical area of participants while still preserving anonymity, the province in which the participants lived was asked instead of their address. To further protect identities, it was possible to choose not to specify a gender.

The cover letter of the questionnaire informed the participants that the data would be used only for scientific purposes, that the raw data would be archived for a maximum of five years, and they would be accessible only by researchers at the University of Modena and Reggio Emilia. Google Forms was used only to spread the questionnaire and to collect data. All the collected data were then downloaded and stored on a professional computer and protected by a password.

It was possible to complete the questionnaire only after the participants declared that they understood the methods and purposes of the study by clicking on the “I give the consent” option in reference to the processing of personal data. Participants who disagreed were redirected to a thank you message.

## 3. Results

Table 1 and Table 2 describe the overall characteristics of caregivers and children of our study involving 218 families living in the provinces of Modena (70%—153 families) and Reggio Emilia (30%—65 families), in the Emilia-Romagna region, with children aged from 6 months to 3 years.

**Table 1 healthcare-12-00453-t001:** Demographic and socio-economic characteristics of caregivers and families, overall and stratified by age categories.

Variables		Total (%)	6 m–1 y	2–3 y	*p*-Value
**Provinces**					<0.001
	Modena	153 (70%)	58 (88%)	95 (63%)	
	Reggio Emilia	65 (30%)	8 (12%)	57 (38%)	
**Median Age—Mother**		34 (32–37)	33 (31–36)	35 (32–38)	0.035
**Median Age—Father**		36 (33–40)	36 (32–40)	36 (33–40)	0.3
	Unknown	1	0	1	
**Nationality—Mother**					0.4
	Italian	212 (97%)	63 (95%)	149 (98%)	
	Not Italian	6 (3%)	3 (5%)	3 (2%)	
**Nationality—Father**					0.4
	Italian	210 (96%)	65 (98%)	145 (95%)	
	Not Italian	8 (4%)	1 (2%)	7 (2%)	
**Educational Level—Mother**					0.2
	Up to high school	71 (33%)	24 (36%)	47 (31%)	
	University	146 (67%)	42 (64%)	104 (69%)	
	Unknown	1	0	1	
**Educational Level—Father**					0.7
	Up to high school	114 (56%)	35 (53%)	79 (57%)	
	University	91 (44%)	31 (47%)	60 (43%)	
	Unknown	13	0	13	
**Occupational Category—Mother**					0.7
	Upper class	66 (30%)	23 (35%)	43 (28%)	
	Middle class	42 (19%)	11 (17%)	31 (20%)	
	Working class	100 (46%)	30 (45%)	70 (46%)	
	Not working	1 (1%)	2 (3%)	8 (5%)	
**Occupational Category—Father**					0.5
	Upper class	64 (31%)	24 (37%)	40 (29%)	
	Middle class	52 (25%)	17 (26%)	35 (25%)	
	Working class	88 (43%)	24 (37%)	64 (46%)	
	Not working	1 (1%)	0 (0%)	1 (1%)	
	Unknown	13	1	12	
**Marital Status**					0.9
	Married	106 (49%)	31 (47%)	75 (49%)	
	Living with partner	105 (48%)	33 (50%)	72 (47%)	
	Single/Separated Widowed	7 (3%)	2 (3%)	5 (3%)	
**Family Income**					0.4
	EUR > 2500	110 (50%)	32 (48%)	78 (51%)	
	EUR 1500–2499	52 (24%)	13 (20%)	39 (26%)	
	EUR < 1499	11 (5%)	3 (5%)	8 (5%)	
	Unknown	45 (21%)	18 (27%)	27 (18%)	
**Number of Children**					<0.001
	1 child	157 (72%)	61 (94%)	96 (63%)	
	2 children	57 (26%)	4 (6%)	53 (35%)	
	3 or more children	3 (1%)	0 (0%)	3 (2%)	
	Unknown	1	1	0	
**Grandparents’ Aid**					0.3
	Yes	159 (73%)	51 (77%)	108 (71%)	
	No	59 (27%)	15 (23%)	44 (29%)	
**Other Aids**					<0.9
	Babysitter	20 (9%)	6 (9%)	14 (9%)	
	Domestic worker	44 (20%)	12 (18%)	32 (21%)	
	Both aids	5 (2%)	1 (2%)	4 (3%)	
	None	149 (68%)	47 (71%)	102 (67%)	

**Table 2 healthcare-12-00453-t002:** Characteristics of children in our sample, overall and stratified by age categories.

		Tot = 218	6 m–1 y	2–3 y	*p*-Value
**Overall**			66 (30%)	152 (70%)	
**Sex**					0.7
	Male	108 (50%)	32 (48%)	78 (51%)	
	Female	110 (50%)	34 (52%)	74 (49%)	
**Chronic Condition**					0.015
	Yes	4 (2%)	0 (0%)	4 (3%)	
	No	213 (98%)	65 (100%)	148 (97%)	
	Unknown	1	1	0	
**Nursing**					0.2
	Breastfeeding	138 (63%)	40 (61%)	98 (64%)	
	Combination	51 (23%)	20 (30%)	31 (20%)	
	Formula	29 (14%)	6 (9%)	23 (15%)	
	Breastfeeding duration mean	12 (9–18) months	12 (10–12) months	13 (8–18) months	
**Weaning**					0.030
	Conventional	77 (35%)	15 (23%)	62 (41%)	
	Supplementary feeding	115 (53%)	43 (65%)	72 (47%)	
	Baby-led	26 (12%)	8 (12%)	18 (12%)	
	Unknown	1	0	1	
	Weaning start mean	6 (5.5–6) months	6 (5.5–6) months	6 (5.5–6) months	
**Baby food**					0.9
	Daily	70 (32%)	20 (30%)	50 (33%)	
	1–2 times per week	47 (22%)	15 (23%)	62 (41%)	
	1–2 times per month	32 (15%)	11 (17%)	21 (14%)	
	Not used	69 (31%)	20 (30%)	49 (32%)	
**Meals**					<0.001
	At home	50 (23%)	40 (61%)	10 (6.6%)	
	At home and school	168 (77%)	26 (39%)	142 (93%)	

The questionnaire was filled out by 208 mothers and 10 fathers. The median age for the mothers was 34 years old (interquartile range 32–37 years), and the median age for the fathers was 36 years old (interquartile range 33–40 years). Most of the parents were Italian. For marital status, 49% of the parents were married, 48% were living with a partner, and the remaining parents were either single, separated, or widowed (respectively, 2%, 1%, and 1%).

In terms of educational levels, 67% of the mothers held a master’s degree or a bachelor’s degree, and 33% had a secondary school diploma or a professional school diploma. Among the fathers, 44% held a master’s or a bachelor’s degree, 56% had a secondary school diploma, and 13 did not respond.

Regarding occupational categories, mothers were distributed as follows: 30% upper class, 19% middle class, 46% working class, and 5% not working. Conversely, fathers are distributed as 31% upper class, 25% middle class, 43% working class, 1% not working, and for 13 no answer was given. The families in our sample had a monthly family income as follows: 5% less than EUR 1499, 24% between EUR 1500 and 2499, and 50% above EUR 2500. The remaining 21% preferred not to respond.

Out of the families in our sample, 72% had one child, 26% had two children, and the remaining 3 had three or more children. One participant did not respond to the question. In 73% of cases, our families could rely on the assistance of their grandparents, while 27% did not. Additionally, or in lieu of being helped by grandparents, 9% of families relied on a babysitter for childcare, 20% on a domestic worker for household duties, 2% had both types of assistance, and the remaining 68% did not rely on any form of assistance [30].

In our sample, 92% of families had no members following a personalized diet, while in the remaining, 8% were vegetarian, had celiac disease, were lactose intolerant, pescatarian, vegan, or flexitarian, and 4 did not provide a response (Table 1).

### 3.1. Children’s Characteristics

Regarding the children of our sample, we had 50% females and 50% males, respectively, 110 and 108. In total, 66 children (30%) were aged between 6 months and 1 year (6 to 23 months), and 152 children (70%) were aged between 2 and 3 years (24 to 47 months). They were mostly healthy and had no food allergies or intolerances.

Regarding the nursing characteristics of the children in our sample, 63% of the children were breastfed exclusively, 23% of the children were nursed in combination, and 14% were fed formula solely. The interquartile range for the length of breastfeeding was 9 to 18 months, with a median of 12 months. Weaning usually started at 6 months of age (median), with an interquartile range of 5 to 6 months. In terms of weaning approaches, 35% of parents opted for conventional weaning, 53% chose supplementary feeding, and 12% opted for baby-led weaning. During the weaning process, baby food was used daily in 32% of cases, not utilized in 31% of cases, used 1–2 times per week in 25% of cases, and used at least 1–2 times per month in 15% of cases (Table 2).

### 3.2. Eating Habits

The frequency of the main food groups intake for the entire sample and adherence to CREA guidelines [16], stratified by age category, is presented in Table 3 and Table 4, respectively. For the food frequencies of all the food items (e.g., red meat, white meat, etc.) please refer to Appendix A.

Regarding cereals consumption, it is observed that 67% of our pediatric population consumed one portion of cereals multiple times a day, predominantly pasta and bread.

Concerning total meat consumption, 17% of the sample adhered to the CREA Guidelines (GL), consuming meat three times a week, while 63% consumed meat more frequently, ranging from four times a week to several times a day. In 38% of cases, red meat consumption exceeded the once per week recommendation. For fish, 49% of the sample ate it two to three times a week, and 40% ate it at least once. As for legumes, 50% of the children consumed two to three servings a week, aligning with the GL and adhering to recommendations for both the age group of 6 months to 1 year (46%) and 2–3 years (53%).

Fifty-seven percent of children consumed more cheese than advised by the GL, with a higher prevalence in children aged 6 months to 1 year than 2–3 years old (65% and 53%, respectively).

Regarding vegetable consumption, up to 50% of children adhered to the GL, consuming them more than once a day, with the percentage decreasing from 73% in the age group 6 months–1 year and to 39% from 2 to 3 years.

Approximately 13% of children consumed vegetables less than four times a week. Regarding fruit, 56% of children adhered to the GL, consuming more portions per day, with a higher percentage in the 6 months–1 year age group, reaching 67%. Concerning eggs, 76% consumed less than the GL.

As for adherence to the optimal consumption of sweets, soft drinks, and processed food, we observed an age-dependent declining trend with children having an extremely high adherence, over 88% in early infancy, decreasing up to 51% in children aged 2–3 years old.

Overall adherence to the GL was higher in children aged 6 months (7.00) to 1 year than 2–3 years old (6.00).

To evaluate the impact of schools on adherence to dietary guidelines (GL), we incorporated the food consumption data of a typical school weekly meal for the 77% of children who consumed lunch at school.

Figure 1 shows how having lunch at school increased the overall number of food categories to which the child adhered. That being said, eating a school lunch demonstrated varying effects on GL adherence, improving it in certain categories while worsening it in others.

Figure 2 (refer to Appendix A for frequencies) depicts the variations in food categories when accounting for school meals among children having lunch at school. The inclusion of school meals notably increased meat and cheese consumption. Overall, on one hand, meat and cheese consumption exceeded the GL in 89% and 83% of children, respectively. On the other hand, counting school meals increased the adherence to vegetables, fruit, fish, eggs, and legumes recommendations, resulting in a general higher adherence score in children who had meals at school (median of 8.00 vs. 7.00).

Table 5 shows the difference in GL adherence among children who ate at home and at school.

The results of the clustering analysis of food categories adherence showed two distinct clusters of adherence phenotypes (Figure 3). Food categories that more frequently clustered were adherence to the GL on industrial products, sweets, soft drinks, and cereals (forming a sub-cluster) and legumes, fish, vegetables, and fruit (forming a second sub-cluster) with a Jaccard similarity index (which defines the co-occurrence of a pair of food categories) varying from 0.62 to 0.76 for the first sub-cluster and from 0.35 to 0.54 for the second sub-cluster. This cluster was followed by a second cluster formed by the remaining food categories showing a Jaccard similarity index ranging from 0.17 to 0.34 and forming a much looser association. When accounting for school meals, three clusters seemed to appear, with one resembling the first cluster. A second cluster formed by adequate adherence to eggs, fish, cow’s milk, and dried fruit that branches from the same cluster of the first one and an independent loose cluster of meat, yogurt, and cheese (please refer to Appendix A for Jaccard index matrices).

Finally, we asked parents if and how often their pediatricians spoke to them about a healthy diet and environmental sustainability. Most parents reported that their pediatricians discussed healthy eating at each visit (41%), followed by often (36%), sporadically (19%), and never (4%). Interestingly, families whose pediatricians talked about a healthy diet often or at each visit had a higher adherence score (median of 6 vs. 7). However, the majority of pediatricians never addressed topics related to environmental sustainability (78%), with only 7% performing this often or at each visit.

## 4. Discussion

The study encompassed 218 families with children aged 6 months to 3 years in the Modena and Reggio Emilia provinces. The findings revealed parents’ higher education levels and diverse occupational statuses. Breastfeeding was prevalent (63%), and varied weaning practices were observed.

In our population, we have observed some unhealthy dietary patterns; in particular, most children exceeded the recommended meat and cheese intake while insufficiently consuming vegetables, fruit, and legumes. Furthermore, consumption of vegetables and fruits declined with increasing age categories. Conversely, eating sweets, soft drinks, and processed food increased with age. School meals notably impacted dietary compliance, elevating meat and cheese intake above the recommended GL while improving vegetable, fruit, fish, eggs, and legume intake.

In our population, we have identified noteworthy unhealthy dietary patterns. Specifically, a significant proportion of children exhibited elevated consumption levels of meat and cheese, exceeding the recommended GL, coupled with a suboptimal intake of essential food groups such as vegetables, fruits, and legumes. Notably, the consumption of vegetables and fruits exhibited a declining trend as age categories advanced. Conversely, there was an observed increase in the consumption of sweets, soft drinks, and processed foods as age advanced.

The influence of school meals on dietary habits was evident, resulting in heightened adherence to meat and cheese consumption, albeit exceeding recommended thresholds, and concurrently enhancing the intake of vegetables, fruits, fish, eggs, and legumes. Through clustering analysis, distinct dietary patterns emerged, underscoring both prevalent adherence to unhealthy food groups and commendable compliance with healthier dietary choices.

Our findings align with those reported by Steenbergen et al. in 2021 [31], who conducted a study on the dietary habits of Dutch children aged 1 to 3 years. The outcomes of our study parallel theirs, revealing similar dietary patterns. Notably, the intake of animal proteins in the Dutch population was generally within acceptable limits, with only 5% above the recommended guidelines. However, vegetable consumption fell below recommended levels in over 50% of toddlers, despite each child consuming at least one serving of vegetables daily.

Focusing on vegetable consumption in our population, up to 50% of children adhered to the recommended GL, with adherence rates decreasing from 73% in the age group of 6 months to 1 year, and further declining to 39% in the 2 to 3 years age bracket. Additionally, in the Dutch study it was found that the consumption of bread, cereals, and soft drinks exceeded the recommended guidelines. In contrast to this, our population exhibited an exceptionally high adherence rate in early infancy, exceeding 88%, which gradually decreased to 51% in children aged 2 to 3 years.

In 2014, Tognon et al. conducted a study focusing on adherence to the Mediterranean diet in European children aged 2 to 9 years [32]. The analysis considered the daily intake of vegetables and legumes, fresh and dried fruits, cereals and potatoes, fish, meat, and dairy, using data from SACINA. It is important to note that, unlike our study, this covered a different age range with older children, and the socio-economic heterogeneity across various European countries must also be considered.

In this study, Swedish children were found to have below-standard consumption levels for both meat and dairy. Both Swedish and Estonian children exhibited higher intakes of cereals and potatoes compared to established guidelines. Optimal fish consumption was observed in Spanish children. Similar to our analyses, this study also assessed how the introduction of school meals influences adherence to the Mediterranean diet guidelines. In Swedish preschoolers (aged 2 to 5 years), the addition of school meals resulted in decreased adherence, while the opposite trend was observed in school-aged children (aged 6 to 9 years), leading to improved adherence. Improved adherence with the addition of school meals (both in preschool and school age) was also noted in Hungarian and Spanish children. In our population, having lunch at school demonstrated varying effects on GL adherence, improving it in certain categories (vegetables, fruit, fish, eggs, and legumes) while worsening it in others (meat and cheese).

Through an examination of international scientific literature, it becomes evident that most studies exploring the dietary habits of young children tend to concentrate on these habits and their implications for the child’s growth. They also investigate the potential consequences of an unbalanced dietary choice. However, these studies often overlook the crucial connection between a healthy diet and environmental sustainability. Our study aims to fill this gap by emphasizing the importance of addressing both aspects, shedding light on their interconnected nature. Notably, pediatricians in the Italian healthcare system often discussed healthy eating topics but rarely addressed issues related to environmental sustainability.

This study emphasizes the need for targeted interventions to improve children’s dietary habits, especially in light of the inconsistent changes observed in the adherence to healthy diets, as well as infants’ well-being during and beyond pandemic times [33,34]. Greater attention should be given to proper nutrition in the early years of life, extending through the preschool-age period.

According to Temme et al., adopting a healthy diet would also help to reduce the environmental impact by lowering greenhouse gas emissions and minimizing the extent of cultivated land for animal fodder, in order to promote both sustainability and the reduction in environmental damage [35].

In addressing the issue of children eating more meat and dairy products than recommended, it is crucial to consider the potential health and sustainability implications of such dietary patterns. Research has shown that reducing the intake of animal proteins during infancy may also decrease the risk of experiencing an early resurgence of obesity [1,36].

In pursuit of this goal, the EAT-Lancet dietary GL recommend decreasing the intake of animal proteins, particularly those derived from red meat, milk, and dairy products, transitioning towards increased consumption of plant-based foods [37]. Limiting the intake of animal protein, with particular reference to animal-origin proteins, GL suggests higher quantities of poultry meat over beef and pork. Likewise, there is a need to enhance the consumption of legumes, nuts, and vegetables, and in addition, refined grains should be replaced with whole grains whenever feasible [11].

The many sustainable diets, such as the Mediterranean diet and the EAT-Lancet diet, could yield benefits in terms of both health outcomes and reduction in greenhouse gas emissions and land use. It has the potential to contribute to climate change mitigation, align with other planetary health goals, and to help to lower the incidence of diet-related diseases [11]. This is due to greenhouse gas emissions being reduced by up to 31% and arable land use decreasing by up to 42% [37]. While the EAT-Lancet diet may meet the recommended intake for most nutrients, it is important to note that its nutritional adequacy has not been extensively studied yet, and reliable and objective data are needed, as assessed by an index [38]. This is significant for understanding the potential impact of transitioning to a more sustainable food consumption pattern on nutrient intakes, especially in the early stages of life, since in toddlers, even a minor modification in dietary habits can significantly influence dietary benefits and prove to be a successful approach, but it can also lead to nutritional inadequacies.

This study underlines the importance of the school system in helping our children achieve the proper intake for important food categories, such as fruit and vegetables, whose consumption significantly improves, reaching recommended levels, with the addition of the school meal. Regarding the role of school meals in increasing meat and cheese consumption, the authors emphasize the importance of not altering the school menu but, instead, raising parental awareness and suggesting a reduction in the number of home meals involving these products. For many children, school meals constitute their sole access to animal protein, highlighting the significance of this food category in their diet. Schools could instill a strong foundation of knowledge about healthy food choices and sustainable eating practices, thereby empowering children and their families to make informed decisions about their diets, creating a holistic approach to promote not only individual health but also contributing to the well-being of the planet we all share.

Our study also highlights that an unhealthy lifestyle pattern can manifest from the early years of a child’s life, and this emphasizes how crucial it is to begin early intervention programs to promote a healthy lifestyle and guarantee lasting benefits for the children’s health in the future. Furthermore, the environmental impact of very early life nutrition, infant formula, baby food consumption, and the kind of weaning has spiked the interest of researchers in recent years. Several studies acknowledge the environmental footprint of baby food in particular formula feeding, indicating breastfeeding as pivotal in sustainable diets [39,40,41]. Townsend and Pitchford observed how baby-led weaning promoted toddlers’ liking for healthy and sustainable foods such as carbohydrates [42].

The goal is to establish a diet that is both health-promoting and environmentally sustainable. To this end, pediatricians play a central role in conveying the correct messages to parents about the importance of proper and balanced nutrition for the development and growth of the child while also keeping an eye on environmental sustainability and the environmental impact of nutrition. Undoubtedly, given the insufficient free time of pediatricians during regular visits, public health officials might play a relevant role in addressing these topics. Promoting healthy eating habits also requires adopting caregivers’ perspectives and implementing nutritional education in households and school food programs, and it should encompass both the physical and psychological factors of healthy and appropriate eating and nutrition [43]. Furthermore, studies have reported a connection between children’s eating habits and mothers’ dissatisfaction with their toddlers’ body size, highlighting the significance of parental influence on their nutritional needs [44].

The results of this study must be read considering its limitations. First, the questionnaire was written and distributed in Italian, which implied a high prevalence of Italian respondents. Most respondents come from high-income families with a high educational status. Furthermore, to preserve anonymity, the lack of sensitive data did not allow us to dig deep into the sociocultural and health-related aspects linked to correct nutrition or sensibility to environmental sustainability. Additionally, to maintain questionnaire anonymity and brevity, we were unable to determine the precise school meal consumption of children, even when accounting for school meal offerings. This presents a significant limitation only in evaluating the nutritional aspects of a child’s healthy meal. The results of our study suggested that the issue must be thoroughly studied, enriching the sample to cover a higher percentage of respondents and performing a more accurate assessment of children’s dietary habits, including school meal consumption.

This being said, our data shed light on the impact of children’s nutrition not only on their own health but also on our planet’s health. Furthermore, the data account for the role played by educational institutions in correcting children’s nutrition. Comprehending the complex connection between the dietary choices of children and the wider environmental consequences is essential for shaping sustainable eating habits.

Creating new public health policies aimed at safeguarding children’s health requires a comprehensive nutritional assessment of infants and toddlers, extending its scope to encompass the family, school, and community. We could achieve improvement by raising awareness among families about balancing the proper intake of food categories, as for the GL. Supporting an integrated approach that combines social and educational initiatives is crucial and future research should prioritize fostering sustainable eating habits within communities to facilitate the transformation of dietary habits and encourage healthier lifestyles.

## 5. Conclusions

Our study underscores the crucial role of children’s dietary habits in their own and the environment’s sustainability, emphasizing the need to achieve recommended levels of essential food categories like fruits and vegetables, while reducing meat and dairy consumption. It highlights the importance of school meals’ offering in achieving adherence, while underlining the necessity of raising parental awareness about integrating said meals with meals offered at home. Furthermore, our findings show early signs of unhealthy lifestyle patterns in children, underscoring the urgency of early intervention programs for lifelong health commitment. The study advocates for a diet promoting both individual health and environmental sustainability. In conclusion, a comprehensive approach is essential to instill healthy eating habits early, addressing individual well-being and the broader environmental impact. This necessitates collaborative efforts from healthcare professionals, educators, and policymakers to encourage positive changes in children’s dietary behaviors and contribute to sustainable lifestyles in communities.

## Figures and Tables

**Figure 1 healthcare-12-00453-f001:**
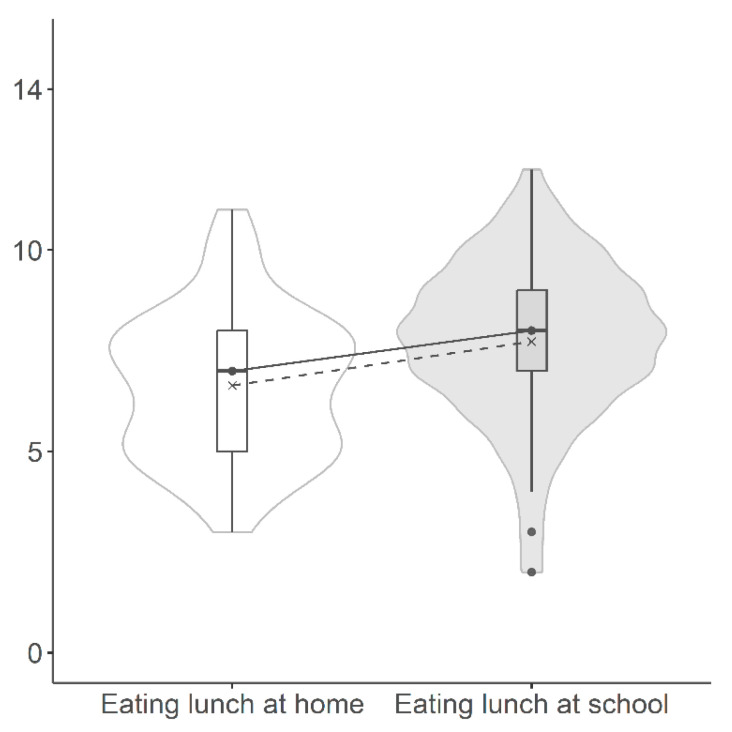
Adherence score to food consumption recommended by the Italian guidelines for healthy eating (CREA 2018) [15] differentiated by school meal consumption. Black continuous lines connect median values (black dot), while black dashed lines connect means (x).

**Figure 2 healthcare-12-00453-f002:**
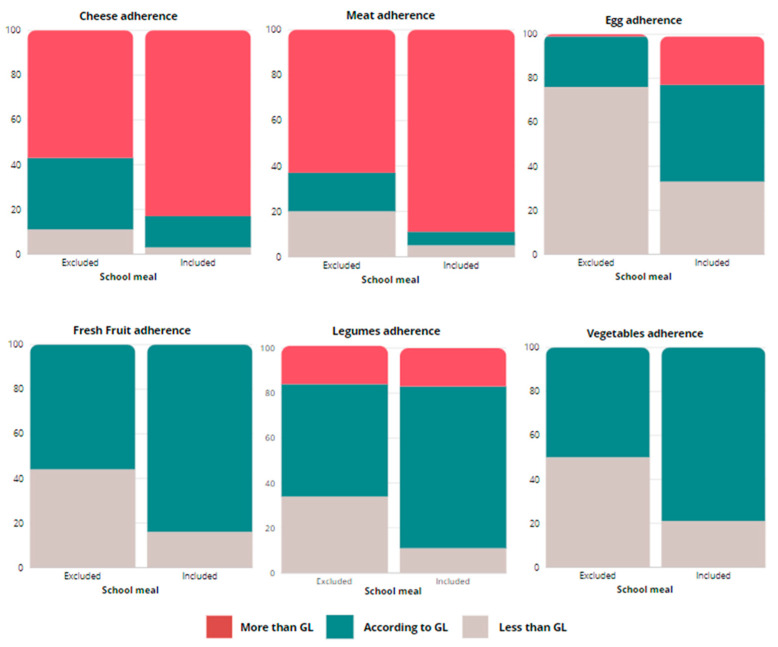
Differences for adherence in food categories comparing food categories excluding and including standard school meals for the children who had lunch at school.

**Figure 3 healthcare-12-00453-f003:**
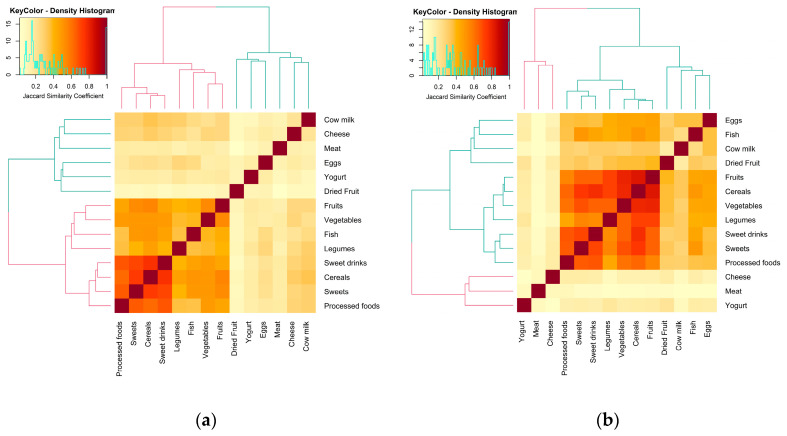
Dendrogram and heatmap describing adherence clustering for main food categories and representing food categories adherence co-occurrence. Co-occurrence was calculated using the Jaccard similarity index, where 0 (light yellow) represents when two food categories adherence never occur together and 1 (dark red) if they always appear together. Clustering was obtained using the complete linkage agglomerative hierarchical clustering method: (**a**) without accounting for school meals, (**b**) accounting for school meals.

**Table 3 healthcare-12-00453-t003:** Children’s food frequency consumption for the main food categories in the overall sample and divided by age categories.

		Tot = 218	6 m–1 y = 66	2–3 yrs = 152	*p*-Value
**Total Cereals**					0.004
	2–3 per day	147 (67%)	53 (80%)	94 (62%)	
	1 per day	29 (13%)	10 (15%)	19 (13%)	
	Almost daily	35 (16%)	3 (4.5%)	32 (21%)	
	3–5 per week	5 (2%)	0 (0%)	5 (3%)	
	2 per week	2 (1%)	0 (0%)	2 (1%)	
**Total Meat**					0.2
	1–2 per day	28 (13%)	12 (18%)	16 (11%)	
	Almost daily	31 (14%)	10 (15%)	21 (14%)	
	4 per week	79 (36%)	23 (35%)	56 (37%)	
	3 per week	37 (17%)	10 (15%)	27 (18%)	
	2 per week	30 (14%)	6 (9%)	24 (16%)	
	1 per week	7 (3%)	1 (2%)	6 (4%)	
	Rarely/never	28 (13%)	12 (18%)	16 (11%)	
**Fish**					0.5
	4–5 per week	8 (4%)	3 (5%)	5 (3%)	
	2–3 per week	107 (49%)	37 (56%)	70 (46%)	
	1 per week	88 (40%)	22 (33%)	66 (43%)	
	Rarely/never	15 (7%)	4 (6%)	11 (7%)	
**Cheese**					0.12
	1 per day	77 (35%)	32 (48%)	45 (30%)	
	Almost daily	34 (16%)	9 (14%)	25 (16%)	
	4–5 per week	13 (6%)	2 (3%)	11 (7%)	
	3–4 per week	42 (19%)	8 (12%)	34 (22%)	
	2–3 per week	27 (12%)	7 (11%)	20 (13%)	
	1–2 per week	25 (11%)	8 (12%)	17 (11%)	
**Legumes**					0.001
	Almost daily	11 (5%)	6 (9%)	5 (3%)	
	4–5 per week	25 (11%)	15 (23%)	10 (7%)	
	2–3 per week	108 (50%)	28 (42%)	80 (53%)	
	1 per week	58 (27%)	11 (17%)	47 (31%)	
	Rarely/never	16 (7%)	6 (9%)	10 (7%)	
**Vegetables**					<0.001
	1–2 per day	108 (50%)	48 (73%)	60 (39%)	
	1 per day	39 (18%)	5 (8%)	34 (22%)	
	4–6 per week	42 (19%)	10 (15%)	32 (21%)	
	2–3 per week	20 (9%)	2 (3%)	18 (12%)	
	1 per week	5 (2%)	0 (0%)	5 (3%)	
	Rarely/never	4 (2%)	1 (2%)	3 (2%)	
**Fruit**					0.001
	1–2 per day	121 (56%)	44 (67%)	77 (51%)	
	1 per day	49 (22%)	9 (14%)	40 (26%)	
	4–6 per week	29 (13%)	8 (12%)	21 (14%)	
	2–3 per week	10 (5%)	2 (3%)	8 (5%)	
	1 per week	4 (2%)	0 (0%)	4 (3%)	
	Rarely/never	5 (2%)	3 (5%)	2 (1%)	
**Sweets**					<0.001
	1–2 per day	3 (1%)	0 (0%)	3 (2%)	
	Almost daily	21 (10%)	1 (2%)	20 (13%)	
	3–5 per week	33 (15%)	6 (9%)	27 (18%)	
	1–2 per week	71 (33%)	6 (9%)	65 (43%)	
	Rarely/never	90 (41%)	53 (80%)	37 (24%)	
**Processed Foods**					<0.001
	1–2 per day	3 (1%)	0 (0%)	3 (2%)	
	3–5 per week	18 (9%)	1 (2%)	17 (12%)	
	1–2 per week	55 (26%)	7 (11%)	48 (33%)	
	Rarely/never	136 (64%)	58 (88%)	78 (53%)	
	Unknown	6	0	6	

**Table 4 healthcare-12-00453-t004:** Adherence to food consumption recommended by the Italian guidelines for healthy eating (CREA 2018) [16] for the main food categories calculated only on home-consumed meals, overall and stratified by age categories. * Expressed as the median (IQR).

		Tot = 218 (%)	6 m–1 y = 66	2–3 yrs = 152	*p*-Value
**Overall Adherence score ***		7.00 (5.00, 8.00)	7.00 (6.00, 8.00)	6.00 (5.00, 7.25)	0.015
**Cereals Adherence**					0.10
	Less than GL	7 (3%)	0 (0%)	7 (5%)	
	According to GL	211 (97%)	66 (100%)	145 (95%)	
**Meat Adherence**					0.6
	Less than GL	43 (20%)	11 (17%)	32 (21%)	
	According to GL	37 (17%)	10 (15%)	27 (18%)	
	More than GL	138 (63%)	45 (68%)	93 (61%)	
**Fish Adherence**					0.3
	Less than GL	103 (47%)	26 (39%)	77 (51%)	
	According to GL	107 (49%)	37 (56%)	70 (46%)	
	More than GL	8 (4%)	3 (5%)	5 (3%)	
**Legumes Adherence**					<0.001
	Less than GL	74 (34%)	17 (26%)	57 (38%)	
	According to GL	108 (50%)	28 (42%)	80 (53%)	
	More than GL	36 (17%)	21 (32%)	15 (10%)	
**Cheeses Adherence**					0.2
	Less than GL	25 (11%)	8 (12%)	17 (11%)	
	According to GL	69 (32%)	15 (23%)	54 (36%)	
	More than GL	124 (57%)	43 (65%)	81 (53%)	
**Vegetables Adherence**					<0.001
	Less than GL	110 (50%)	18 (27%)	92 (61%)	
	According to GL	108 (50%)	48 (73%)	60 (39%)	
**Fresh Fruit Adherence**					0.029
	Less than GL	97 (44%)	22 (33%)	76 (49%)	
	According to GL	121 (56%)	44 (67%)	77 (51%)	
**Dried Fruit Adherence**					0.095
	Less than GL	136 (62%)	48 (73%)	88 (58%)	
	According to GL	22 (10%)	6 (9%)	16 (11%)	
	More than GL	60 (28%)	12 (18%)	48 (32%)	
**Milk Adherence**					<0.001
	Less than GL	136 (62%)	57 (86%)	79 (52%)	
	According to GL	72 (33%)	9 (14%)	63 (41%)	
	More than GL	10 (5%)	0 (0%)	10 (7%)	
**Yogurt Adherence**					0.001
	Less than GL	158 (72%)	57 (86%)	101 (66%)	
	According to GL	37 (17%)	9 (14%)	28 (18%)	
	More than GL	23 (11%)	0 (0%)	23 (15%)	
**Eggs Adherence**					0.3
	Less than GL	165 (76%)	54 (82%)	111 (73%)	
	According to GL	50 (23%)	11 (17%)	39 (26%)	
	More than GL	3 (1%)	1 (2%)	2 (1%)	
**Sweets Adherence**					<0.001
	According to GL	161 (74%)	59 (89%)	102 (67%)	
	More than GL	57 (26%)	7 (11%)	50 (33%)	
**Soft Drinks Adherence**					<0.001
	According to GL	166 (76%)	61 (92%)	105 (69%)	
	More than GL	52 (24%)	5 (8%)	47 (31%)	
**Processed Food Adherence**					<0.001
	According to GL	136 (62%)	58 (88%)	78 (51%)	
	More than GL	82 (38%)	8 (12%)	74 (49%)	

**Table 5 healthcare-12-00453-t005:** Adherence to food consumption recommended by the Italian guidelines for healthy eating (CREA 2018) [16] for the main food categories calculated including standard school meals for those children who had lunch at school, overall and stratified by consuming lunch at home or at school. * Expressed as the median (IQR).

		Tot = 218 (%)	Eating Lunch at School (168)	Eating Lunch at Home (50)	*p*-Value
**Overall Adherence Score ***		8.00 (6.00–9.00)	8.00 (7.00, 9.00)	7.00 (5.00, 8.00)	<0.001
**Cereals Adherence**					0.052
	Less than GL	2 (1%)	0 (0%)	2 (4%)	
	According to GL	216 (99%)	168 (100%)	48 (96%)	
**Meat Adherence**					<0.001
	Less than GL	11 (5%)	2 (1%)	9 (18%)	
	According to GL	14 (6%)	6 (9%)	8 (16%)	
	More than GL	193 (89%)	160 (95%)	33 (66%)	
**Fish Adherence**					0.5
	Less than GL	103 (47%)	78 (46%)	25 (50%)	
	According to GL	107 (49%)	85 (51%)	22 (44%)	
	More than GL	8 (4%)	5 (3%)	3 (6%)	
**Legumes Adherence**					<0.001
	Less than GL	25 (11%)	11 (7%)	14 (28%)	
	According to GL	157 (72%)	140 (83%)	17 (34%)	
	More than GL	36 (17%)	17 (10%)	19 (38%)	
**Cheeses Adherence**					<0.001
	Less than GL	7 (3%)	0 (0%)	7 (14%)	
	According to GL	30 (14%)	18 (11%)	12 (24%)	
	More than GL	181 (83%)	150 (89%)	31 (62%)	
**Vegetables Adherence**					0.002
	Less than GL	45 (21%)	27 (16%)	18 (36%)	
	According to GL	173 (79%)	141 (84%)	32 (64%)	
**Fresh Fruit Adherence**					<0.001
	Less than GL	35 (16%)	16 (10%)	19 (38%)	
	According to GL	183 (84%)	152 (90%)	31 (62%)	
**Dried Fruit Adherence**					0.8
	Less than GL	136 (62%)	103 (61%)	33 (66%)	
	According to GL	22 (10%)	18 (11%)	4 (8%)	
	More than GL	60 (28%)	47 (28%)	13 (26%)	
**Milk Adherence**					0.001
	Less than GL	136 (62%)	94 (56%)	42 (84%)	
	According to GL	72 (33%)	65 (39%)	7 (14%)	
	More than GL	10 (5%)	9 (5)	1 (2%)	
**Yogurt Adherence**					0.3
	Less than GL	158 (72%)	118 (70%)	40 (80%)	
	According to GL	37 (17%)	30 (18%)	7 (14%)	
	More than GL	23 (11%)	20 (12%)	3 (6%)	
**Eggs Adherence**					<0.001
	Less than GL	73 (33%)	29 (17%)	44 (88%)	
	According to GL	97 (44%)	92 (55%)	5 (10%)	
	More than GL	48 (22%)	47 (28%)	1 (2%)	
**Sweets Adherence**					0.026
	According to GL	161 (74%)	59 (89%)	102 (67%)	
	More than GL	57 (26%)	7 (11%)	50 (33%)	
**Soft Drinks Adherence**					0.063
	According to GL	166 (76%)	123 (73%)	43 (86%)	
	More than GL	52 (24%)	45 (27%)	7 (14%)	
**Processed Food Adherence**					0.003
	Less than GL	82 (38%)	72 (43%)	10 (20%)	
	More than GL	136 (62%)	96 (57%)	40 (80%)	

## Data Availability

The article’s data will be shared after a reasonable request to the corresponding author.

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
