# Peer review of "Investigating Eating Habits of Children Aged between 6 Months and 3 Years in the Provinces of Modena and Reggio Emilia: Is Our Kids’ Diet Sustainable for Their and the Planet’s Health?"

_healthcare, 2024, doi:10.3390/healthcare12040453_

Round 1

Reviewer 1 Report

Comments and Suggestions for Authors

Remove contraction - Line 56

There needs to be a better explanation of the dietary assessment (including an explanation of why previously validated Mediterranean diet indices were not used - Lines116-129

There are spelling and grammatical errors in the manuscript.

2.2

For international audiences, it would be helpful to define the food frequency categories, specifically what differentiates red vs. white meat, and which foods are considered cereals (i.e., processed sugary grains vs. whole grains). Are pasta and bread considered cereals? Are whole grain and processed grains all in the same category?  What are “other cereals?” Which foods are in the “processed foods” category?

Please provide a better description of the consumption frequencies assessed from school meals.  Was there an assumption that children consumed all items provided or did the schools assess individual children’s consumption?  Giving a child a healthy lunch does not necessarily equate to consumption of that lunch.

Section 3.2 is extremely difficult to follow – needs to be edited with phrases like “in regards to” and “concerning…” removed.

Results

While the descriptive statistics are helpful, it would be more impactful to conduct inferential statistics related to breastfeeding (and other variables) and the dietary patterns.

Lines 299-302, 369-371: These statements are only valid if actual school lunch consumption (not the school menu) was assessed.  If actual consumption was not measured, this seems to be more of an analysis of school lunch offerings. Please clarify as school menu assessment is a significant limitation to this study.

Comments on the Quality of English Language

Minimal errors in grammar

Author Response

The authors would like to thank the reviewers for taking the time to review this manuscript. We are deeply thankful to both reviewers for their valuable suggestions which helped us improve the quality and clarity of our manuscript.  Please find the detailed responses below and the corresponding revision which can also be found in track change in the re-submitted files.

Reviewer 2 Report

Comments and Suggestions for Authors

In general, the article is very interesting and well structured, it touches upon very important aspects of the children health, nutrition and planet condition. Nevertheless, I have a few comments:

Abstract

Lines 31-34 - these two sentences refer to consumption without school meals - I know this because I read the article, but this is not clear from Abstract. Please clarify.

Introduction

Line 78 – actual world population is 8 bln (UN 2022)

Line 98 – which country? (in Abstract also)

Line 99 – in my opinion there in mistake – not “assumption” but “consumption”

Methods

Table 1 – “occupational category” - on which criteria were people assigned to a given category / answer? What is different between classes? Who did decided about it - participants? It is not clear. I found information on this in Results (lines 212-213 - European Classification of Socioeconomic Groups) – but it should be described in Methods.

Results

Point 3.2 - statistically significant differences between age groups are not described.

What is the total/overall adherence to nutritional guidelines? Results are presented for the main food categories, but I think authors should also give adherence of whole diet – if the method used allows it. Maybe Figure 1 presents this, but it was not described enough.

Discussion

The authors did not compare their results to the work of other authors. The authors mentioned that there are not a lot of such studies, but they are available, including foreign ones. Maybe it should be compared with older age groups. In my opinion, own results are not discussed enough.

Author Response

(The authors gave the same response as above.)
